# Pathological Approach to Kidney Allograft Infection

**DOI:** 10.3390/biomedicines11071902

**Published:** 2023-07-05

**Authors:** Suwasin Udomkarnjananun, Kroonpong Iampenkhae

**Affiliations:** 1Division of Nephrology, Department of Medicine, Faculty of Medicine, Chulalongkorn Univeristy and King Chulalongkorn Memorial Hospital, Thai Red Cross Society, Bangkok 10330, Thailand; 2Excellence Center for Organ Transplantation (ECOT), King Chulalongkorn Memorial Hospital, Thai Red Cross Society, Bangkok 10330, Thailand; 3Renal Immunology and Transplantation Research Unit, Faculty of Medicine, Chulalongkorn University, Bangkok 10330, Thailand; 4Center of Excellence in Translational Research in Inflammation and Immunology, Faculty of Medicine, Chulalongkorn University, Bangkok 10330, Thailand; 5Department of Pathology, Faculty of Medicine, Chulalongkorn University and King Chulalongkorn Memorial Hospital, Thai Red Cross Society, Bangkok 10330, Thailand; kroonpong@gmail.com

**Keywords:** adenovirus, BK virus, cytomegalovirus, JC virus, kidney allograft infection, pyelonephritis

## Abstract

Infectious agents can pose a significant challenge in kidney transplantation, as they have the potential to cause direct infections in the transplanted kidney. These infections can lead to a decline in kidney function and reduce the longevity of the transplanted kidney. Common post-transplant allograft infections include bacterial pyelonephritis and the BK virus infection, while adenovirus, JC virus, and cytomegalovirus are less frequent but can also lead to significant allograft dysfunctions. The histopathological features of these infections are characterized by the infiltration of inflammatory cells in the kidney interstitial area and the presence of viral nuclear inclusions or cytopathic changes in the renal tubular epithelial cells. The confirmation of causative organisms can be achieved by immunohistochemical staining or the visualization of viral particles using electron microscopic examination. However, these methods typically require a longer turnaround time and are not readily available in developing countries, unlike standard hematoxylin-eosin staining. Notably, the differential diagnosis of interstitial inflammation in kidney allografts almost always includes T cell-mediated rejection, which has a different treatment approach than allograft infections. The aim of this review was to prompt clinicians to identify diverse pathological alterations as observed in kidney allograft biopsies, thereby facilitating further investigations and the management of suspected kidney allograft infections.

## 1. Introduction

Kidney transplantation is considered the most effective treatment option for patients suffering from end-stage kidney disease. Compared to dialysis, kidney transplant recipients generally experience improved quality of life and lower mortality rates [1]. However, not all transplant recipients are able to achieve a smooth and uncomplicated post-transplantation course, as infectious complications are common and can contribute significantly to kidney allograft loss and mortality [2,3,4]. The degree of immunosuppressive therapy utilized can be directly correlated with the incidence of post-transplantation infections [3,5].

An infection in kidney transplant recipients may manifest as a systemic infection or may involve specific organs that are similar to non-transplant patients [5]. In the first post-transplant month, infections can result from either donor-derived or hospital-acquired sources. The reactivation of latent viruses can be commonly observed during the first six months following transplantation. Beyond the first year of transplantation, the risk of infection decreases, at the level of immunosuppression is usually lowered. Nonetheless, transplant recipients remain immunosuppressed and, therefore, remain susceptible to infections. Microorganisms that can directly infect the kidney allograft pose a significant threat as they can severely damage the transplanted kidney, diminish its function, and curtail the longevity of the allograft [2]. Infections in the kidney allograft can result in the infiltration of inflammatory cells in the interstitium and renal tubular epithelial cells, which can cause allograft dysfunction, leading to irreversible interstitial fibrosis and tubular atrophy [6,7,8,9,10]. This review aims to elucidate the various patterns of interstitial infiltration caused by common post-kidney transplantation infections and offer insights into differentiating them from other conditions that may also cause interstitial inflammation. Additionally, the current state of knowledge and management approaches for each infectious disease could be briefly reviewed.

In this article, histopathology images were derived from formalin-fixed paraffin-embedded (FFPE) tissues, with a renal tissue section thickness ranging from 2 to 3 µm. For an electron microscopic examination, 3% glutaraldehyde was utilized for tissue fixation. These methodologies strictly adhered to well-established protocols and guidelines, as outlined in textbooks and the scientific literature [11,12].

## 2. Bacterial Pyelonephritis

Bacterial urinary tract infections (UTIs) are common in kidney transplant recipients, with a prevalence ranging from 20 to 70% after transplantation [13,14]. Allograft pyelonephritis can be defined as the presence of systemic symptoms (fever, hemodynamic instability, or leukocytosis) in addition to allograft pain or urinary tract symptoms [15]. The established risk factors of a bacterial UTI include female sex, advanced age, structural abnormalities, diabetes mellitus, bladder dysfunction, and prolonged stent or catheter placement [13,14,15,16]. The dosage of immunosuppression, including anti-rejection treatments, has also been shown to be associated with a post-transplant UTI. Recently, a body mass index (BMI) above 25 kg/m^2^ has been identified as another risk for a post-transplant UTI, which could be related to immune dysfunctions caused by excessive adipose tissue and higher weight-based dosages of immunosuppressive medications in these populations [10]. The etiological agents of a UTI following kidney transplantation commonly involve Gram-negative bacteria, including *Escherichia coli*, *Klebsiella* spp., *Enterococcus* spp., *Proteus* spp., or *Pseudomonas aeruginosa*. Additionally, due to the hospitalization and urinary tract catheterization associated with kidney transplant surgery, Gram-positive bacteria such as *Staphylococcus saprophyticus*, *Staphylococcus epidermidis*, or *Staphylococcus aureus* may also be implicated, albeit less frequently compared to Gram-negative bacteria [14,17]. It is worth noting that the contemporary use of broad-spectrum antibiotics has led to growing concerns regarding the prevalence of multi-drug resistant (MDR) organisms [18,19]. Moreover, heightened immunosuppression levels, particularly in terms of tacrolimus concentration and corticosteroid dosage, have exhibited a positive correlation with the increased incidence of UTIs caused by MDR organisms [10].

The importance of UTIs after kidney transplantation lies in their negative impact on allograft outcomes. Studies including mixed immunosuppressive regimens from different transplant eras have demonstrated that kidney transplant recipients who experienced UTIs have had an inferior allograft function and are at higher risk of death or allograft loss compared with non-UTI recipients [20,21,22,23,24,25]. Even in the current modern era of immunosuppression, recipients who experienced recurrent UTIs had a worsened allograft function, and those experiencing UTIs within the first transplant month had significantly lower patient survival compared with recipients who did not have UTIs [10]. Recipients with post-kidney transplant UTIs, particularly allograft pyelonephritis, should receive an appropriate evaluation to prevent a recurrence, including the ultrasonography of the transplant kidney, prostate examination, revisiting voiding history and hygiene care, and reviewing current immunosuppressive medications. Tissue injury from sexual intercourse and hormonal changes during the peri-menstruation period can also predispose to UTIs. In high-risk recipients, residual urine evaluation and urodynamic study are necessary to reveal the hidden cause of UTIs [15].

The classic pathology of allograft pyelonephritis is characterized by neutrophilic tubulitis and interstitial inflammation [26]. This includes the presence of polymorphonuclear cells (PMNs) in the renal interstitium and tubular basement membrane (Figure 1). PMNs can also accumulate in the renal tubular lumens, causing a PMN cast, infiltrating the peritubular capillary areas, and forming microabscesses. It should be noted that PMNs can also be presented in the interstitium or tubular lumens in other conditions besides pyelonephritis, such as acute rejection, acute glomerulonephritis, or acute tubular necrosis. However, these conditions usually have only a few PMNs compared to the large numbers of PMNs that are present in acute allograft pyelonephritis. Moreover, the inflammatory areas in allograft pyelonephritis are patchy or zonal, compared with more diffuse cellular infiltrations found in acute T cell-mediated rejection (TCMR) [27,28].

The treatment of allograft pyelonephritis includes proper antibiotics and supportive care as needed. The prevention of a recurrent UTI is also crucial when minimizing its impact on allograft function. Based on the results from randomized controlled trials, treating asymptomatic bacteriuria within the first two months or after two months of transplantation did not translate to any clinical benefits compared to untreated asymptomatic bacteriuria [29,30].

## 3. BK and JC Polyomavirus Infection

The BK virus (BKV) is a circular, double-stranded DNA virus that measures 40 nm in diameter and lacks an envelope. The first reported incidence of BKV infection in kidney transplantation occurred in Sudanese kidney transplant recipients who exhibited ureteric stenosis in 1971 [31]. Primary asymptomatic infection can occur during early childhood, potentially through the respiratory route, and may persist latently in the renal tubular epithelial cells and epithelial cells of the urogenital tract. The seropositivity of BKV exceeds 90% in children after 10 years old [32,33,34]. After kidney transplantation, 10–20% of recipients experience BK viremia, and 1–10% of recipients may progress to BKV-associated nephropathy (BKVAN), which can lead to permanent damage, including transplanted kidney and the loss of kidney function [32]. Both the reactivation of BKV in transplant recipients and donor-derived BKV can lead to active BKV infections after transplantation [35,36,37]. The risk factors for BKV infection after kidney transplantation include the degree of immunosuppression (particularly with thymoglobulin and high-dose steroids), delayed graft function, a higher degree of the human leukocyte antigen (HLA) mismatch, young or elderly recipients, male sex, ureteral stent placement, and deceased donor kidney transplantation [33,38]. BKV seropositivity could be used to predict BK viremia and BKVAN, with the highest risk observed in cases of donor-positive to recipient-negative serology [39]. Nonetheless, cellular immunity played a primary role in suppressing BKV infection [40]. A meta-analysis of the BKV-specific enzyme-linked immunosorbent spot (ELISPOT) assay revealed that BK viremic kidney transplant recipients with a positive BKV-specific ELISPOT assay were more likely to clear the virus. On the other hand, BK viremic recipients without a T cell response against BKV antigens were unable to clear the virus and faced a higher risk of developing active ongoing BK viremia and BKVAN [41].

The main histopathologic change that has been observed in BKVAN is the infiltration of mononuclear cells in the kidney interstitium and tubular epithelial cells. This inflammatory response triggered by the virus can lead to allograft injury and permanent kidney damage [42]. Furthermore, viral nuclear inclusion changes in renal tubular epithelial cells are commonly observed in the inflamed tubular area (Figure 2A), which present in various forms, including enlarged dense basophilic smudgy nuclear inclusions, finely granular amphophilic to basophilic nuclear inclusions with basophilic peripheral chromatin ring, a ground-glass appearance with irregular central clearing, or the large vesicular appearance of nuclei [27,28]. The key diagnostic criterion is positive Simian Virus 40 (SV40) immunohistochemical staining in the infected nuclei (Figure 2B), which can be helpful in mild or early cases of BKVAN when no typical nuclear inclusions are observed in standard staining. The earliest cytopathic changes in BKVAN were observed within the medullary parenchyma of the kidney, which could be missed if the biopsy core contained only the cortical tissue [43]. Similar to other allograft infectious diseases, the areas of inflammation from BKVAN were typically patchy in contrast to the diffuse infiltration observed in TCMR. BKV particles can be observed in the nuclei of renal tubular epithelial cells under an electron microscopic examination, demonstrating their lattice arrangement in a 40 nm virion (Figure 3).

Currently, two main classifications of BKVAN are available: the 2013 American Society of Transplantation (AST) and the 2018 Polyomavirus Nephropathy (PVN) Banff classification [43,44]. A recent study showed that the 2018 Banff classification (Table 1) performed slightly better when predicting the prognosis of BKVAN [45]. In the validation study for the 2018 PVN Banff classification, the 2 years allograft failure rates in recipients with class 1, class 2, and class 3 PVN were 3%, 9%, and 18%, respectively [46].

Atypical histological presentations of the BKV infection in the kidney allograft were reported, including the crescentic formation resulting from BKV infection in the visceral and parietal epithelial cells of Bowman’s capsule [47,48]. A concurrent acute rejection should be suspected if endarteritis, fibrinoid vascular necrosis, or glomerulitis are observed [49]. Additionally, urothelial hyperplasia and carcinoma have been reported to be associated with BKV, as the polyomavirus is a carcinogenic virus (poly-; many, -oma; tumor) [50].

In recipients with persistent BK viremia that exceeds 3 log_10_ copies/mL within 3 weeks (probable BKVAN), an increase to more than 4 log_10_ copies/mL (presumptive BKVAN) or a biopsy-proven BKVAN, immunosuppression reduction is the primary treatment modality [49]. If recipients do not respond to immunosuppressive reduction, second-line treatments such as switching mycophenolic acid (MPA) to the mammalian target of the rapamycin inhibitor (mTORi) or leflunomide may be necessary, although evidence supporting these therapies is currently lacking [33,49].

The human pathogen JC polyomavirus (JCV) can also cause an allograft dysfunction similar to BKV. JCV infects over 80% of the adult population and can silently reside in the urinary tract epithelium [51]. In immunosuppressed kidney transplant recipients, JCV can be reactivated, resulting in JCV-associated nephropathy. The true incidence of JCV-associated nephropathy is unknown and usually occurs at a later post-transplantation period compared to BKV, which mostly presents within the first-year post-transplant [9,52]. JCV-associated nephropathy cannot be distinguished from BKVAN based on histopathology since they both exhibit similar viral inclusions, cytopathic changes, and positive SV40 immunohistochemical staining and have the same virion size (40 nm). However, JCV-associated nephropathy is more likely to have interstitial fibrosis and tubular atrophy (IFTA), which could be related to its late presentation after transplantation [9]. To diagnose JCV-associated nephropathy, the histology should be compatible with the aforementioned pathology observed in BKVAN, with undetectable BK viremia. The polymerase chain reaction (PCR) for JCV should indicate replication in the urine and/or blood of suspected transplant recipients. Most cases of JCV-associated nephropathy respond well to immunosuppressive reduction [9,52].

## 4. Adenovirus Nephritis

Adenovirus is a non-enveloped, double-stranded DNA virus [53,54]. More than 80% of the adult population are latently infected, where some serotypes are capable of infecting the genitourinary tract [54]. Solid organ transplant recipients are at risk of developing adenovirus infection from either the reactivation of a latent virus or a donor-derived infection. Among adult solid organ transplantation, the highest incidence of adenovirus infection is in intestinal, lung, and kidney transplantation [54]. Although the adenovirus infection in kidney transplantation is not uncommon (with an incidence ranging from 4–6%), adenovirus nephritis is rare, and its true incidence is unknown (less than 0.5%) [53,55]. A recent case series of 11 kidney transplant recipients with adenovirus nephritis indicated that adenovirus nephritis was more common in males and in recipients of deceased donor kidney transplantation [6]. Other risk factors included the use of the anti-lymphocyte antibody, donor-positive to recipient-negative serostatus and pediatric transplantation [53,55]. Unlike BKV infection, fever is a common presentation of adenovirus nephritis and is often accompanied by either gross or microscopic hematuria [6]. Adenovirus can also manifest as hemorrhagic cystitis, pneumonia, hepatitis, enterocolitis, or a disseminated infection [54].

The pathological features of adenovirus nephritis include marked tubular necrosis and interstitial inflammation [27,28]. Tubular necrosis can be severe and illustrates tubular basement membrane disruption, frank tubular destruction, and mimics cortical necrosis [53]. Hemorrhagic interstitial nephritis can also be observed with granulomatous formation. In the inflamed necrotic tubules, enlarged, smudgy, basophilic viral nuclear inclusion is presented (Figure 4A), which is similar to those found in BKVAN. A recent study indicated that adenovirus nephritis had less interstitial fibrosis compared to BKVAN, possibly due to the increased transcripts of the host immune response inhibition in the kidney allograft tissue of recipients with adenovirus nephritis [6]. The confirmation of adenovirus nephritis involved positive immunohistochemical staining against adenovirus in the affected renal tubular epithelial cells (Figure 4B). Compared to BKV, electron microscopic examination of adenovirus showed a larger virion at approximately 70 to 80 nm [53].

Adenovirus nephritis rarely results in allograft failure by itself because most of the cases respond well to immunosuppressive reduction. Subsequent allograft failure is often associated with acute rejection episodes after a decrease in immunosuppressive medications [6]. Second-line treatments for those who do not achieve viral clearance after a reduction in immunosuppression include intravenous immunoglobulin (IVIg), cidofovir, brincidofovir, and T-cell adoptive therapy [54,55].

## 5. Cytomegalovirus Infection

Cytomegalovirus (CMV) is a double-stranded DNA virus that can establish lifelong latency within the host. This virus can infect various cell types, including endothelial cells, epithelial cells, and leukocytes [56]. In immunosuppressed transplant recipients, CMV can reactivate or directly infect the host from the donor during organ transplantation. The risk factors of CMV infection (detectable CMV replication) and CMV disease (CMV infection with attributable symptoms) post-kidney transplantation include donor-positive to recipient-negative serostatus, the use of lymphocyte-depleting agents, net immunosuppression, acute rejection episodes, and older donor [57,58,59]. A recently randomized controlled TRANSFORM trial demonstrated that the use of de novo everolimus with a calcineurin inhibitor (CNI) was associated with a lower incidence of CMV infection compared to standard CNI with the MPA regimen [60]. The CMV infection in kidney transplant recipients can manifest as asymptomatic viremia, leading to systemic infection with leukopenia and fever or organ-specific diseases such as pneumonitis and colitis. Moreover, CMV infection has been reported to be associated with an inferior allograft function, allograft rejection, decreased patient survival, and decreased allograft survival [59,61,62,63]. To prevent post-transplant CMV infection, a CMV prevention strategy must be applied to every kidney transplant recipient, either through prophylaxis or a preemptive strategy. CMV prophylaxis is recommended in a donor-positive to recipient-negative serostatus transplantation, recipients who received lymphocyte-depleting agents, or recipients with a human immunodeficiency virus (HIV) infection [57,64]. This strategy involves the empirical administration of the prophylactic dose anti-viral therapy to recipients in the first 3–6 months after transplantation. On the other hand, a preemptive strategy monitors for CMV viremia by PCR and treats with a standard anti-viral dose once the pre-specified threshold of CMV viremia was met (usually 1500 to 3000 copies/mL). In the current recommendation from the International Consensus, either prophylaxis or a preemptive strategy can be used in kidney transplant recipients [57].

CMV is commonly presented as a viremia or systemic infection, and direct infection in the kidney allograft is less common than BKV infection [8,65,66]. In the case of CMV-associated nephropathy, pathology shows allograft interstitial inflammation and tubulitis, predominantly with mononuclear cell infiltration (Figure 5) [65]. Three main differences can be observed that help differentiate CMV nephritis from BKVAN [53]. First, the typical viral nuclear inclusions in CMV infection show a peri-nuclear halo or owl’s eye-type inclusion. Second, viral inclusion can be present in both the nuclei and cytoplasm of the renal tubular epithelial cells with the enlargement of both the nuclei and cytoplasm of tubular epithelial cells (cytomegaly). Lastly, CMV has the ability to infect endothelial cells, resulting in a glomerulitis-like pathology that is not present in cases of BKVAN. The confirmation of CMV tubulo-glomerulitis can be made by positive immunohistochemical staining in the affected tubules and glomeruli. The bull’s eye appearance of the viral particle (150–200 nm), with a dense core and thick capsule, can be observed from electron microscopic examination [8].

The treatment of CMV allograft nephritis includes anti-viral medication, with the most commonly used being intravenous ganciclovir and oral valganciclovir. Both medications can be used for non-life-threatening CMV disease. However, intravenous ganciclovir can be preferred as an initial treatment of life-threatening CMV disease when the oral bioavailability of valganciclovir is uncertain [57]. The adjustment of immunosuppressive medications should be made based on the immunologic risk of the recipients. Drug-resistant CMV infection should be suspected when there is persistent or recurrent CMV viremia or CMV disease during prolonged anti-viral therapy that lasts more than two weeks at a full dose. The treatment options for ganciclovir/valganciclovir-resistant CMV infection include foscarnet, cidofovir, or adoptive T-cell therapy, according to the current recommendation [57,58].

## 6. Other Differential Diagnoses of Kidney Allograft Interstitial Cell Infiltration

In addition to bacterial pyelonephritis and viral infection in the kidney allograft, the differential diagnoses of interstitial cell infiltration in the kidney allograft consisted of TCMR, drug-induced interstitial nephritis, and post-transplant lymphoproliferative disease (PTLD). 

Clinical and subclinical TCMR have been shown to lead to compromised allograft function and reduced allograft survival [67,68,69,70]. TCMR can be classified into acute and chronic active TCMR according to the Banff 2019 classification [71]. Acute TCMR grade IA and IB require at least 25% interstitial inflammation of non-sclerotic cortical parenchyma (i2 or i3), while chronic active TCMR grade IA and IB requires interstitial inflammation involving over 25% of sclerotic parenchyma (i-IFTA2 or i-IFTA3) and more than 25% of the total cortical parenchyma (ti2 or ti3). The primary components of infiltrated cells in TCMR include cytotoxic (CD8+) T cells, helper (CD4+) T cells, and monocytes [72,73,74]. B lymphocytes typically constitute less than 10% of infiltrating cells; however, plasma cell-rich infiltrates can be linked with significant interstitial edema and the poor outcome of cellular rejection [75]. Infiltrated cell types cannot differentiate TCMR from viral infection morphologically. Nevertheless, the absence of viral nuclear inclusions and/or cytopathic changes in the inflamed tubulointerstitial area could be useful in distinguishing TCMR from allograft viral infection.

Similar to native kidneys, drug-induced interstitial nephritis can also occur in the kidney allograft. However, the incidence of drug-induced interstitial nephritis in the kidney allograft is currently unknown, and its differential diagnosis is more extensive compared to native kidneys because TCMR and allograft infections must also be considered. A high number of eosinophils may suggest drug-induced interstitial nephritis, although they can also be present in TCMR and acute pyelonephritis [27,28,76]. Therefore, the diagnosis of drug-induced interstitial nephritis requires the exclusion of other causes of allograft dysfunction with suspected medications that can commonly cause interstitial nephritis in transplant recipients, such as cotrimoxazole, furosemide, proton pump inhibitors, and acyclovir. The treatment of drug-induced interstitial nephritis typically involves the withdrawal of potentially offending drugs and considers a corticosteroid regimen, which is also an effective treatment for TCMR in case these two conditions cannot be differentiated.

PTLD is an abnormal lymphoproliferation that is associated with the Epstein–Barr virus (EBV) in approximately 50% of cases [77,78,79]. The pathogenesis of EBV-positive PTLD is due to an immunosuppression-related decrease in T cell immune surveillance against EBV-infected cells, which leads to the abnormal proliferation and transformation of these cells. In contrast, EBV-negative PTLD is similar to sporadic lymphoma, which coincidentally occurs in transplant recipients [77,79]. The risk of PTLD is highest in multiorgan and intestinal transplantation, with kidney transplantation having the lowest incidence of PTLD among solid organ transplantation [79]. Other risk factors include the intensity of induction therapy, particularly lymphocyte-depleting agents, which are donor-positive to recipient-negative serostatus, the specific HLA of the recipients (HLA-A26, B18, B21, and B20), and the cumulative immunosuppression dosage [79,80]. Early onset PTLD tends to be EBV-driven and can involve the kidney allograft, with the pathology characterized by monomorphic or polymorphic mononuclear cell infiltration in the kidney interstitium. The main difference from TCMR is the absence of tubulitis and the presence of a space-occupying infiltrate without or minimal tubular cell damage in allograft PTLD [27,28]. The management of PTLD depends on the morphologic classification, which typically includes reducing immunosuppression, rituximab (only in CD20+ PTLD), and chemotherapy [79].

To provide a summary of the approach for evaluating allograft interstitial inflammation or cellular infiltration, differentiation based on the predominant cell type can be employed. This allowed us to distinguish allograft pyelonephritis, which is characterized by abundant PMN infiltration, from drug-induced interstitial nephritis, which is characterized by eosinophil-rich infiltration. When mononuclear cells predominantly infiltrate the pathology, the presence of viral replication detected in serum and identified through immunohistochemical staining can aid in identifying the cause of viral-induced allograft injury. If there is a significant number of mononuclear cell infiltrations with minimal or no tubulitis, the suspicion of PTLD arises, in contrast to TCMR, where prominent tubulitis is observed. A schematic representation of this approach is shown in Figure 6, and a summary of the incidence, clinical presentation, allograft pathology, and treatment is provided in Table 2.

## 7. Conclusions

Allograft infections can lead to a decline in the allograft function and reduced allograft survival. Pathologic features that are commonly observed in allograft infections include interstitial inflammation with various types of cell infiltrations. Acute pyelonephritis can be characterized by a large number of neutrophilic tubulitis and interstitial cell infiltration. Viral infections such as BKVAN, JCV-associated nephropathy, adenovirus nephritis, and CMV nephritis are characterized by the presence of viral nuclear inclusions, which can be detected through immunohistochemical staining. The details of these viral inclusions and surrounding kidney parenchymal changes can provide clues for the specific virus. The differential diagnosis of interstitial cell infiltration in the allograft is presented in Figure 6. It is important for clinicians to be able to identify and manage kidney transplant recipients presenting with allograft dysfunction and interstitial inflammation appropriately. 

## Figures and Tables

**Figure 1 biomedicines-11-01902-f001:**
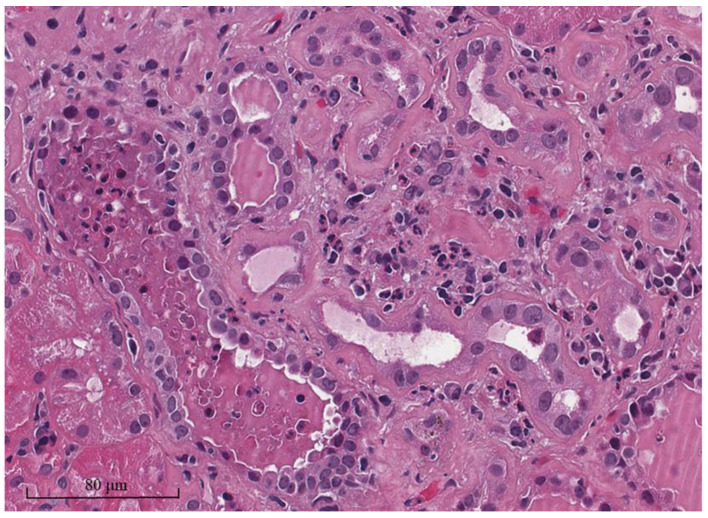
The histopathological examination of allograft pyelonephritis reveals a prominent neutrophilic infiltration in both the interstitial area and peritubular capillaries. The tubular lumen contains a significant number of neutrophils, resulting in the formation of the polymorphonuclear cell cast (hematoxylin and eosin staining). (Authors’ original image).

**Figure 2 biomedicines-11-01902-f002:**
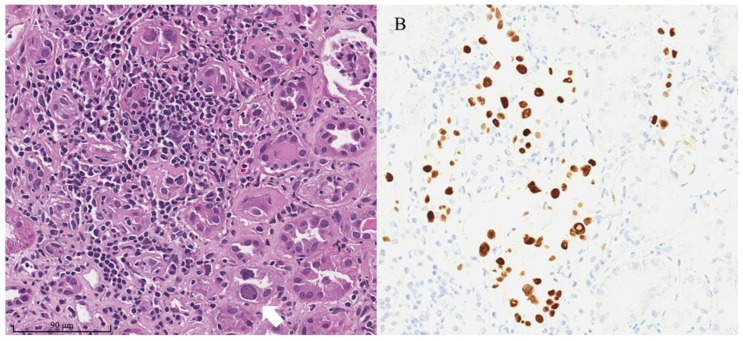
(**A**) Histopathology of BKVAN showing dense mononuclear cells infiltration in the renal interstitium, accompanied by tubulitis (hematoxylin and eosin staining). Viral nuclear inclusion is demonstrated in the affected tissue (arrow). (**B**) Positive SV40 immunohistochemical staining (mouse anti-SV40 monoclonal antibody, ready-to-use) from the same kidney allograft biopsy. (Authors’ original images).

**Figure 3 biomedicines-11-01902-f003:**
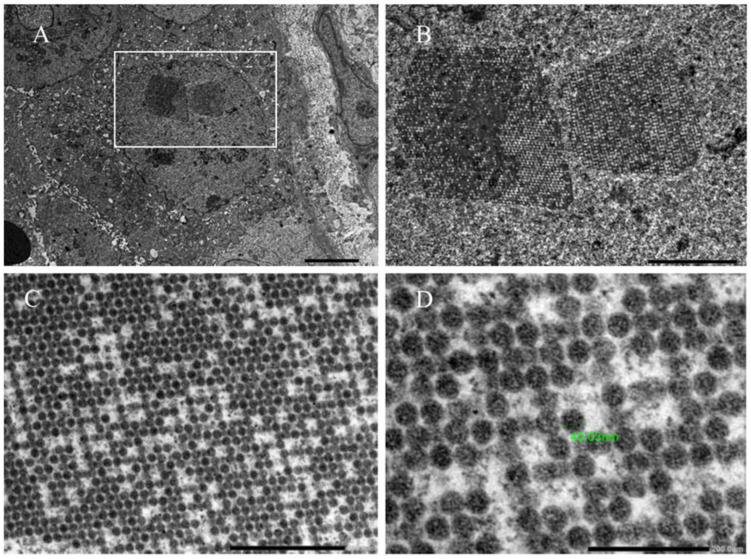
The electron microscopic examination of renal tubular nuclei infected with BKV was performed at various magnifications, ranging from (**A**–**D**), with an increasing resolution of 5 μm, 2 μm, 500 nm, and 200 nm, respectively. White box in (**A**) indicates the area of magnification shown in (**B**). These images were obtained from the same kidney allograft biopsy as shown in Figure 2. Notably, the examination revealed the presence of 40 nm BKV particles (indicated as a green marker in (**D**)) arranged in a lattice-like pattern. This 40 nm viral particle could also be found in the JC virus infection, where immunohistochemical staining could help identify the pathogen. (Authors’ original images).

**Figure 4 biomedicines-11-01902-f004:**
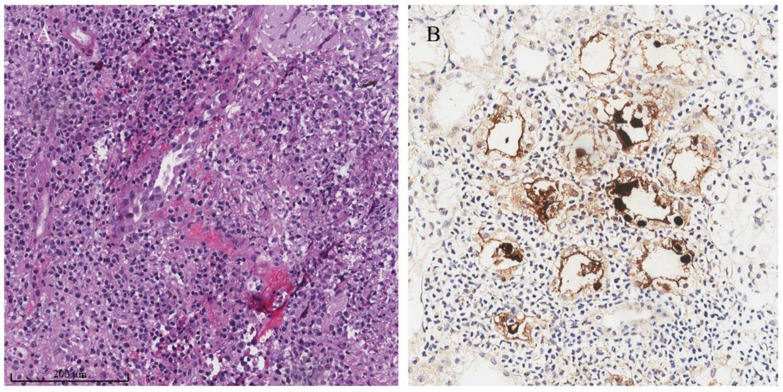
(**A**) The histopathological evaluation of adenovirus allograft nephritis demonstrates severe tubulitis and interstitial inflammation. Early formation of granuloma and interstitial hemorrhage can be observed in the affected tissue (hematoxylin and eosin staining). (**B**) Positive immunohistochemical staining for adenovirus (mouse anti-adenovirus monoclonal antibody, 1:1500) from the same kidney allograft biopsy. (Authors’ original images).

**Figure 5 biomedicines-11-01902-f005:**
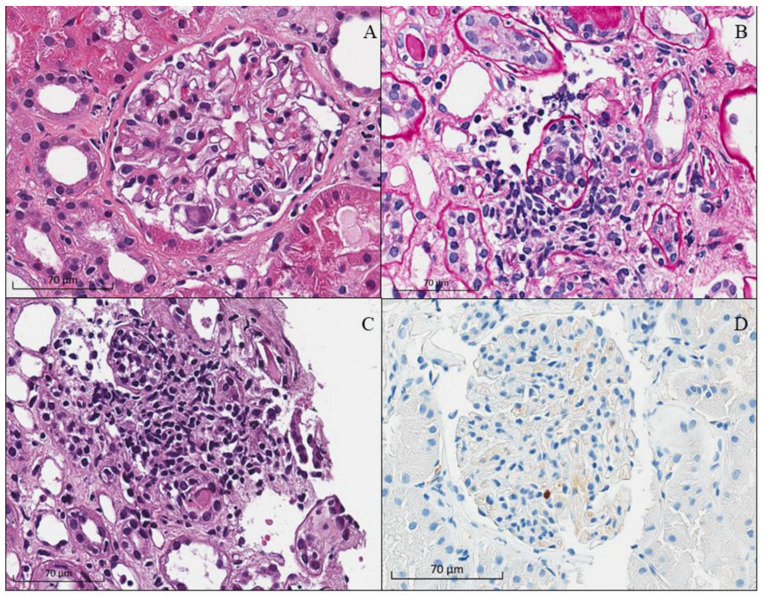
Histopathology of CMV tubulo-glomerulitis in kidney allograft of a kidney transplant recipient showing (**A**) An enlarged intracapillary cell with intranuclear basophilic inclusions (hematoxylin and eosin staining), severe tubulitis with enlarged renal tubular epithelial cell accompanied by interstitial inflammation (**B**) Periodic-acid Schiff to outline the tubular basement membrane, (**C**) The original hematoxylin and eosin staining from which tubulitis is difficult to be distinguished from interstitial inflammation), and (**D**) Positive CMV immunohistochemical staining in the affected glomerulus (mouse anti-CMV monoclonal antibody, ready-to-use) (reproduced with permission from Udomkarnjananun et al., Ref. [65]).

**Figure 6 biomedicines-11-01902-f006:**
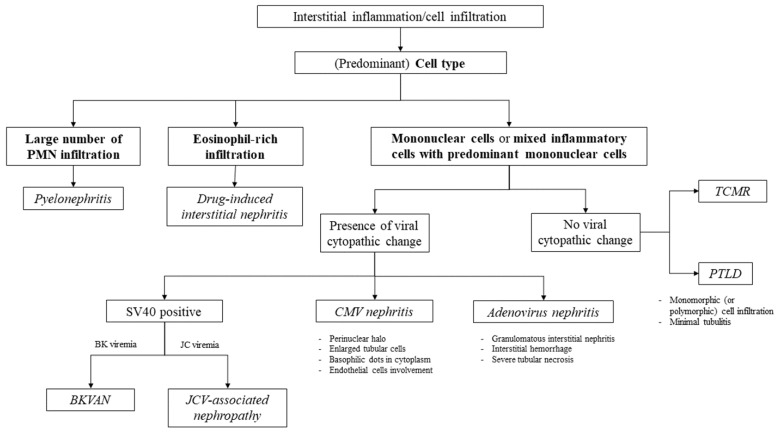
Schematic pathological approach to interstitial inflammation in kidney allograft. BKVAN; BK virus-associated nephropathy, CMV; cytomegalovirus, JCV; JC virus, PMN; polymorphonuclear cells, PTLD; post-transplant lymphoproliferative disorder, TCMR; T cell-mediated rejection, SV40; Simian Virus 40.

**Table 1 biomedicines-11-01902-t001:** The 2018 PVN Banff classification.

Biopsy-Proven BKVAN Class 1		Biopsy-Proven BKVAN Class 2		Biopsy-Proven BKVAN Class 3	
pvl	ci	pvl	ci	pvl	ci
1 (viral replication ≤ 1% of all tubules)	0–1 (interstitial fibrosis ≤ 5% of cortical area)	1 (viral replication ≤ 1% of all tubules)	2–3 (interstitial fibrosis > 25% of cortical area)	-	-
-	-	2 (viral replication > 1% to ≤10% of all tubules)	0–3 (any interstitial fibrosis)	-	-
-	-	3 (viral replication > 10% of all tubules)	0–1 (interstitial fibrosis ≤ 5% of cortical area)	3 (viral replication > 10% of all tubules)	2–3 (interstitial fibrosis > 25% of cortical area)

BKVAN; BK virus-associated nephropathy, ci; interstitial fibrosis, pvl; morphologic polyomavirus load level.

**Table 2 biomedicines-11-01902-t002:** Characteristics of conditions presenting with interstitial inflammation or cellular infiltrations in the kidney allograft [6,9,13,33,53,66,68,76,78,79].

Conditions	Incidence	Clinical Presentation	Kidney Allograft Pathology	Treatment
Bacterial pyelonephritis	20–70% of KTR.	Fever, dysuria, allograft pain, allograft dysfunction.	Neutrophilic tubulitis and interstitial inflammation; PMN casts in tubular lumen.	Antibiotics
BKVAN	1–10% of KTR (BK viremia in 10–20% of KTR).	Allograft dysfunction, usually no systemic symptoms, can present with hematuria or hemorrhagic cystitis.	Mononuclear cell infiltration, viral nuclear inclusion changes (typically enlarged dense basophilic smudgy nuclear inclusions), positive SV40 IHC, 40 nm viral particle from EM.	Immunosuppressive medication reduction is the mainstay treatment. Second-line treatments include switching MPA to mTORi or leflunomide, switching tacrolimus to CsA, IVIG, or cidofovir.
JCV-associated nephropathy	Rare (true incidence is unknown, probably less than 1% of KTR).	Chronic allograft dysfunction, present with later post-transplant period compared to BKVAN.	Indistinguishable from BKVAN; the literature shows higher degree of chronicity in the allograft tissue than in BKVAN, positive SV40 IHC, and 40 nm viral particle from EM.	Immunosuppressive medication reduction is the mainstay treatment.
CMV nephritis	Rare (approximately 0.2% of KTR).	Allograft dysfunction, fever, diarrhea, leukopenia, presence of high level CMV viremia or CMV disease in other organs such as pneumonitis, hepatitis, or colitis.	Mononuclear interstitial inflammation, typical peri-nuclear halo (owl’s eye) inclusion, viral inclusion in both nuclei and cytoplasm, viral cytopathic change in glomeruli (infected endothelial cells), 150–200 nm viral particle from EM.	Ganciclovir or valganciclovir are the first-line treatment, with a reduction in immunosuppressive medications. Foscarnet or cidofovir are used for drug resistant CMV infection.
Adenovirus nephritis	Rare (true incidence is unknown, probably less than 0.5% of KTR).	Allograft dysfunction, fever, gross or microscopic hematuria, can present with systemic involvement such as hemorrhagic cystitis, pneumonia, hepatitis.	Marked tubular necrosis and interstitial inflammation with mononuclear cell infiltration, mimicking cortical necrosis, hemorrhagic interstitial nephritis, granuloma formation, viral nuclear inclusion changes, and 70–80 nm viral particle from EM.	Immunosuppressive medication reduction is the mainstay treatment. Second-line treatments include IVIG or cidofovir.
Drug-induced interstitial nephritis	The true incidence is unknown.	Allograft dysfunction or subclinical inflammation, eosinophilia, or skin rash (not always present), suspected drug such as sulfa, antibiotics, NSAIDs, PPIs, or allopurinol.	High number of eosinophilic interstitial inflammation, requiring the exclusion of other diseases.	Stop potentially offending medications; short-course corticosteroid can be considered.
TCMR	From 5 to 15% (clinical TCMR) to 30% of KTR (including subclinical TCMR) in the first transplant year.	Allograft dysfunction or subclinical inflammation, usually without systemic symptoms.	Mononuclear cell infiltration in the interstitial area with tubulitis or intimal arteritis.	Corticosteroid, anti-thymocyte globulin, intensification of maintenance immunosuppression.
PTLD	5-year cumulative incidence 0.8% 25-year cumulative incidence 3.3%.	Allograft dysfunction, EBV-negative recipient serology (early-onset PTLD tends to be EBV-driven and involves the allograft), unrecognized or prolonged fever, weight loss.	Monomorphic mononuclear cell infiltration in the interstitium, with absence or minimal tubulitis and classification based on the World Health Organization’s classification of PTLD.	Immunosuppressive medication reduction, rituximab in CD20+ PTLD, chemotherapy.

BKVAN; BK virus-associated nephropathy, CMV; cytomegalovirus, CsA; cyclosporine A, EM; electron microscopic examination, IHC; immunohistochemical staining, IVIG; intravenous immunoglobulin, JCV; JC virus, KTR; kidney transplant recipients, MPA; mycophenolic acid, mTORi; mammalian target of rapamycin inhibitor, PTLD; post-transplant lymphoproliferative disorder, SV40; Simian virus 40, TCMR; T cell-mediated rejection.

## Data Availability

All relevant data are provided in this manuscript.

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
