# Peer review of "Pathological Approach to Kidney Allograft Infection"

_biomedicines, 2023, doi:10.3390/biomedicines11071902_

Round 1
Reviewer 1 Report
1. the abstract does not follow a natural structure (introduction, objective…)
2. The images shown throughout the document do not indicate whether they are their own or taken from the articles consulted. In all of them, the type of staining performed, the meaning of differential staining should be explained. In the case of immunohistochemistry, the primary and secondary antibodies used for characterization should be indicated, as well as their dilutions.
3. Fig3D, the authors must indicate the meaning of the green marking in the center of the image. Also, as they show that it is BKV and not another virus.
4. All tables must include a table footer with an explanation of the abbreviations used (Table 1).
5. Fig 6 should be explained before the conclusions.
Minor editing of English language required
Author Response
Reviewer 1:
- The abstract does not follow a natural structure (introduction, objective…).
Authors’ reply: We thank reviewer for the comment. We have adjusted our abstract based on the suggestion from the reviewer and includes the objective of this review. According to the author’s guideline of Biomedicines, the abstract of a review article should be non-structured. Since this is a review article, we did not present the contents in the abstract as in the original article that includes introduction, method, results, and discussion headings.
- The images shown throughout the document do not indicate whether they are their own or taken from the articles consulted. In all of them, the type of staining performed, the meaning of differential staining should be explained. In the case of immunohistochemistry, the primary and secondary antibodies used for characterization should be indicated, as well as their dilutions.
Authors’ reply: We thank the reviewer for the comment. Every image in this manuscript is originally from the Division of Nephrology, Department of Medicine and Department of Pathology, Chulalongkorn University, which are all our own images. The originality of images is added in the figure legends, along with the details of staining.
- Fig3D, the authors must indicate the meaning of the green marking in the center of the image. Also, as they show that it is BKV and not another virus.
Authors’ reply: We thank the reviewer for the comment. The description of the green marker, indication the size of viral particle, is added to the figure legend. The differential diagnosis for the human pathogenic virus that also has 40 nm in size, including JC virus infection, is added to the legend.
- All tables must include a table footer with an explanation of the abbreviations used (Table 1).
Authors’ reply: We thank the reviewer for the suggestion. The description of all abbreviations used in the Table 1 are added in the table footer.
- Fig 6 should be explained before the conclusions.
Authors’ reply: We appreciate the suggestion. The explanation of Figure 6 is added before the conclusions part of the manuscript.
Reviewer 2 Report
Dear Authors,
I have read the manuscript with interest but I think that it is not complete so I send you my comments:
1) please add data of bacterial infections in this clinical situation.
2) please add a table differentiating the viral infections and bacterial infections
3) please add data of pharmacological treatment
Author Response
Reviewer 2:
- Please add data of bacterial infections in this clinical situation.
Authors’ reply: We thank the reviewer for the suggestion. The additional information regarding the bacterial urinary tract infection, including the common pathogenic bacteria, are added in the manuscript.
- Please add a table differentiating the viral infections and bacterial infections.
Authors’ reply: We thank the reviewer for the suggestion. Table 2 is added to the manuscript presenting the summary of the incidence, clinical presentation, allograft pathology, and treatment options for each condition.
- Please add data of pharmacological treatment.
Authors’ reply: Treatment is described in the newly added Table 2.
Reviewer 3 Report
The article submitted for review is interesting.
This is an important article in the context of patient's kidney transplants. It systematizes knowledge about infections in patients after allographic transplants very well. I really like the diagram presented in Fig 6, which systematizes knowledge.
What am I missing? To make the article even more interesting - information on the management and treatment of individual inflammations presented in the form of a graph would be useful. Otherwise, they disappear a bit, and the scheme "breaks off" on the histopathological results. I suggest that there are explanations of abbreviations under Fig 6; then, it will be easier to understand this diagram.
It would also be useful to compare the indicated inflammatory conditions, including the main differences and the frequency of these individual inflammatory complications, e.g., in the form of a table.
I have a question about the figures presented, whether they were the property of the authors. I am thinking here about biopsy pictures.
Minor editing of English language are required
Author Response
Reviewer 3
- To make the article even more interesting - information on the management and treatment of individual inflammations presented in the form of a graph would be useful. Otherwise, they disappear a bit, and the scheme "breaks off" on the histopathological results.
Authors’ reply: We appreciate the reviewer’s suggestion. The information on the treatment of each condition is added in Table 2, along with the short summary of the incidence, clinical presentation, and kidney allograft pathology.
- I suggest that there are explanations of abbreviations under Fig 6; then, it will be easier to understand this diagram.
Authors’ reply: We appreciate the reviewer’s suggestion. The descriptions of abbreviations are added in the Figure legend.
- It would also be useful to compare the indicated inflammatory conditions, including the main differences and the frequency of these individual inflammatory complications, e.g., in the form of a table.
Authors’ reply: The newly added Table 2 includes the difference of clinical characteristics among each inflammatory condition.
- I have a question about the figures presented, whether they were the property of the authors. I am thinking here about biopsy pictures.
Authors’ reply: We thank the reviewer for the comment. All biopsy pathology figures are from our own original images. We added this information in each figure legend.
Round 2
Reviewer 1 Report
Regarding the review carried out by the authors, I still think that if the authors incorporate their own histological material in a bibliographical review, they should indicate the methodology used to obtain the histological images. As the article is stated, we must blindly believe that the markings made with HEosin are those indicated by the authors. How thick are the tissues analyzed? are they frozen or paraffin? What development method do you use? When the histological image is shown, is the immunohistochemical image a serial section of the same biopsy? If not, what's the point of putting them together?
I think exactly the same for electron microscopy images. How do they know it is that virus and not another?
This manuscript has major limitations in its form and content that the authors must review.
Regarding the review carried out by the authors, I still think that if the authors incorporate their own histological material in a bibliographical review, they should indicate the methodology used to obtain the histological images. As the article is stated, we must blindly believe that the markings made with H-Eosin are those indicated by the authors. How thick are the tissues analyzed? are they frozen or paraffin? What development method do you use? When the histological image is shown, is the immunohistochemical image a serial section of the same biopsy? If not, what's the point of putting them together?
I think exactly the same for electron microscopy images. How do they know it is that virus and not another?
This manuscript has major limitations in its form and content that the authors must review.
Author Response
Authors’ reply: We added the information asked by the reviewer in the introduction section. Formalin-fixed paraffin-embedded (FFPE) tissues are universally utilized for renal pathology and are standard practice in our transplant center. The renal tissue sections are 2-3 µm thick. Electron microscopy requires the use of 3% glutaraldehyde for tissue fixation. These protocols adhere to established protocols found in textbooks and literatures and are widely implemented worldwide with minimal variation.
The electron microscopy images in our article (Figure 3) were obtained from the same patient as depicted in Figure 2, who was diagnosed with BK virus-associated nephropathy (BKVAN) and exhibited positive SV40 immunohistochemical staining. The viral particle, measuring 40 nm in size, were observed arranged in a lattice-like pattern, without the presence of other organized deposits. These findings represent a classic and typical case of BKVAN in a kidney transplant recipient.
Considering the primary objective of this article, which is to provide an approach to interpreting interstitial cell inflammation or cellular infiltration in kidney allografts, we primarily focused on highlighting the differences and similarities among viral infections and other conditions in kidney allograft pathology. We aimed to guide clinicians in recognizing these diagnostic clues, particularly in developing countries where access to a comprehensive panel of stains may not be readily available on the same day as H&E staining. Immunohistochemical staining plays a crucial role in differentiating between various viral infections at a later period. By presenting immunohistochemistry images with the H&E staining images that originate from the same allograft biopsies, we aim to provide readers with a comprehensive understanding, even if they initially have access only to H&E staining, which is an important aspect when the clinicians have to consider further management and think about differential diagnoses, and not only for the pathological perspective. The figure legends now include descriptions emphasizing that these immunohistochemistry images and H&E staining images in each figure are from the same biopsies.
We believe that addressing the reviewer’s points and incorporating these additional descriptions has enhanced the clarity of the manuscript and images while maintaining the article’s focus on clinico-pathological correlation. Relevant supporting information on the technical aspects in developing tissue histopathology are included along with the citations for the methods that we used.
Reviewer 2 Report
Dear Authors,
I have read the manuscript and I have not further comments
Author Response
We appreciate the reviewer's comments.
Round 3
Reviewer 1 Report
The manuscript is acceptable in its present form.